# Speculate, then Collaborate:
# Fusing Knowledge of Language Models during Decoding

**Ziyao Wang** [1]  **Muneeza Azmat** [2]  **Ang Li** [1]  **Raya Horesh** [2]  **Mikhail Yurochkin** [2]

## Abstract

Large Language Models (LLMs) often excel in specific domains but fall short in others due to the limitations of their training. Thus, enabling LLMs to solve problems collaboratively by integrating their complementary knowledge promises to improve their performance across domains. To realize this potential, we introduce a novel *Collaborative Speculative Decoding (CoSD)* algorithm that enables efficient LLM knowledge fusion at test time without requiring additional model training. CoSD employs a draft model to generate initial sequences and an easy-to-learn rule or decision tree to decide when to invoke an assistant model to improve these drafts. CoSD not only enhances knowledge fusion but also improves inference efficiency, is transferable across domains and models, and offers greater explainability. Experimental results demonstrate that CoSD improves accuracy by up to 10% across benchmarks compared to existing methods, providing a scalable and effective solution for LLM-based applications. Our code has been released at https://github.com/ATP-1010/CoSD.

## 1. Introduction

State-of-the-art large language models (LLMs), such as GPT-4 (Achiam et al., 2023) and Llama-3 (Dubey et al., 2024), have demonstrated impressive capabilities in generating high-quality text across a variety of domains. These models are trained on vast datasets, allowing them to perform well on a wide range of tasks. However, despite their general effectiveness, no single LLM excels uniformly across all domains. Different models tend to have *comple-*

*mentary knowledge*, with each model specializing in certain areas. For example, one model may be more proficient in technical writing, while another may outperform in creative tasks. This heterogeneity has led to an increasing interest in developing methods that can *fuse the knowledge* of multiple LLMs, enabling users to harness their collective strengths for more robust and versatile applications.

To address these challenges, recent research has shifted focus to *test-time* knowledge fusion, which eliminates the need for retraining by combining model outputs during inference. This approach allows users to leverage the complementary knowledge of LLMs without the training overhead. For example, Wang et al. (2023) proposed a method that selects expert models dynamically at inference time using supervised learning, while Ong et al. (2024) introduced a router model that optimizes the selection of models based on performance and cost. Other approaches focus on integrating outputs through the decoding process, such as token-wise decoding (Shen et al., 2024) and character-wise decoding (Gu et al., 2024), which combine outputs at a fine-grained level. Although these methods offer potential, they often struggle to balance strong knowledge integration with efficiency, which limits their practicality in real-world applications.

In response to these limitations, we propose *Collaborative Speculative Decoding* CoSD, a novel algorithm designed to efficiently fuse the knowledge of multiple LLMs at inference time. CoSD builds upon recent developments in *Speculative Decoding* (Leviathan et al., 2023; Xia et al., 2023) to create an efficient system where multiple LLMs collaborate during the inference process. As shown in Figure 1, CoSD consists of two models: a *draft model* that generates an initial sequence of tokens and an *assistant model* that verifies these tokens in parallel. When the assistant model predicts a token different from that of the draft model, a comparison of their token probabilities is used to determine whether to replace the draft token. This decision-making process can be guided by either a predefined rule set (Rule-Based CoSD) or a pre-trained decision tree (Tree-Based CoSD). The sequence is then regenerated and re-verified iteratively until accepting all tokens, ensuring both accuracy and computational efficiency.

CoSD presents several notable advantages over existing

[1]Department of Electrical and Computer Engineering, University of Maryland, College Park, USA, [2]IBM Research, Yorktown Heights, USA. Correspondence to: Ziyao Wang <ziyaow@umd.edu>.

*Proceedings of the $42^{nd}$ International Conference on Machine Learning*, Vancouver, Canada. PMLR 267, 2025. Copyright 2025 by the author(s).

test-time fusion methods. First, by leveraging speculative decoding, CoSD improves inference efficiency, relying on token probabilities rather than more complex and resource-intensive representations like embeddings or hidden states. Second, CoSD demonstrates superior knowledge fusion due to the carefully designed decision-making process, which can be optimized for specific domains. Third, Rule-Based CoSD is highly transferable across tasks and model pairs; once the rules are established with optimal hyperparameters, they can be applied to a broad range of tasks. Similarly, the decision tree-based approach exhibits strong transferability, even when trained on domain-specific data. Finally, CoSD offers an interpretable framework, i.e., its use of human-readable rules or decision trees provides transparency, making it easier to evaluate, optimize, and understand compared to less transparent deep learning systems.

We validate the effectiveness of CoSD through extensive experiments on standard benchmarks and multiple model pairings. Our results show that CoSD not only significantly enhances the fusion of LLM knowledge but also improves efficiency and transferability across various domains. The key contributions of this work are as follows:

- We introduce CoSD, a novel algorithm that enables efficient test-time fusion of LLM knowledge without requiring retraining.
- CoSD's efficiency and transferability make it practical for a wide range of users, facilitating its implementation through both models and APIs.
- Our experimental results demonstrate that CoSD improves overall accuracy by up to 10% across benchmarks, surpassing the state-of-the-art methods.

## 2. Related Work

**Language Model Fusion** from multiple LMs aims at enhancing the cross-domain performance of the resulting model and reducing bias. The primary efforts for such integration include model merging (Goddard et al., 2024), such as model weight averaging (Wortsman et al., 2022) and linear mode connectivity (Ainsworth et al., 2022; Ito et al., 2024; Wang et al., 2020). Another series of works is called model stacking, which refers to concatenating models along the depth dimension. Wu et al. (2024) and Kim et al. (2023) stack the decoder blocks to expand the depth of Llama models. For large language models, some other research proposes knowledge fusion (Wan et al., 2024). They combine the capabilities of existing LLMs and transfer them into a single LLM. Another important trend of work called Mixture of Expert (MoE) (Zhu et al., 2024; Xue et al., 2024) builds sparse neural networks and only activates a subset of parameters (*i.e.,* experts) for each input. However, these methods either require the fused models to have the same structure or require fine-tuning after fusing to achieve the desired model performance. Towards mitigating these flaws, a new wave of works adopt decoding methods to fuse LMs. Gu et al. (2024) propose a character-wise ensemble decoding method to fuse two LLMs' outputs. Shen et al. (2024) and Wang et al. (2023) fuse model knowledge by training to choose between the generation of different LLMs. In our experiments, we consider several baselines from the latter group of works and observe gains in either efficiency or performance when using our method to merge cross-domain knowledge from different LMs when decoding.

**Speculative Decoding** is an efficient decoding paradigm for LM inference (Xia et al., 2024; Stern et al., 2018; Xia et al., 2023). It accelerates the inference process by first generating draft tokens efficiently, and then using an LLM to verify draft tokens in parallel and correct them if needed (Leviathan et al., 2023), which avoids the autoregression process. In practice, the draft generator in speculative decoding could be a small LM (Chen et al., 2023; Miao et al., 2023; Zhou et al., 2023), a sub-model of an LLM (Zhang et al., 2023; Yang et al., 2023; Elhoushi et al., 2024), or a text database retriever (He et al., 2023; Li et al., 2024). The final generation of speculative decoding will be similar to the autoregressive generation of the target LLM, which is only acceptable when the target LLM has much better performance but is less efficient than the draft generator. No previous work focuses on using speculative decoding to approach the model fusion problem.

## 3. Collaborative Speculative Decoding

In our Collaborative Speculative Decoding system, our purpose is to fuse the predicted sequences of two LLMs efficiently. We define our problem as follows: given an input sequence $x_1, \ldots, x_t$, CoSD uses a draft model $\mathcal{M}_p$ and an assistant model $\mathcal{M}_q$ to collaboratively generate an output sequence $x_{t+1}, \ldots, x_{t+K}$ that integrates both models' knowledge and expertise.

As Figure 1 illustrates, the process begins with the draft model $\mathcal{M}_p$ generating a draft sequence $\widetilde{x}_{t+1}, \ldots, \widetilde{x}_{t+K}$ in an autoregressive manner. Subsequently, the assistant model $\mathcal{M}_q$ verifies the draft tokens by producing an assistant sequence $\hat{x}_{t+1}, \ldots, \hat{x}_{t+K}$. After both sequences are generated, CoSD iterate through the tokens and their corresponding probabilities to verify whether to accept a draft token $\widetilde{x}_{t+i}$ or replace it with the corresponding assistant token $\hat{x}_{t+i}$. Both rule-based or tree-based verification strategies, use token probabilities to determine whether a replacement is necessary. When a replacement occurs, all subsequent draft tokens are discarded, and a new draft sequence is generated starting from the replaced token. This process continues until the output reaches the maximum length or an $< EOS >$ token is generated. The full generation and verification process is elaborated in Algorithm 1 and described

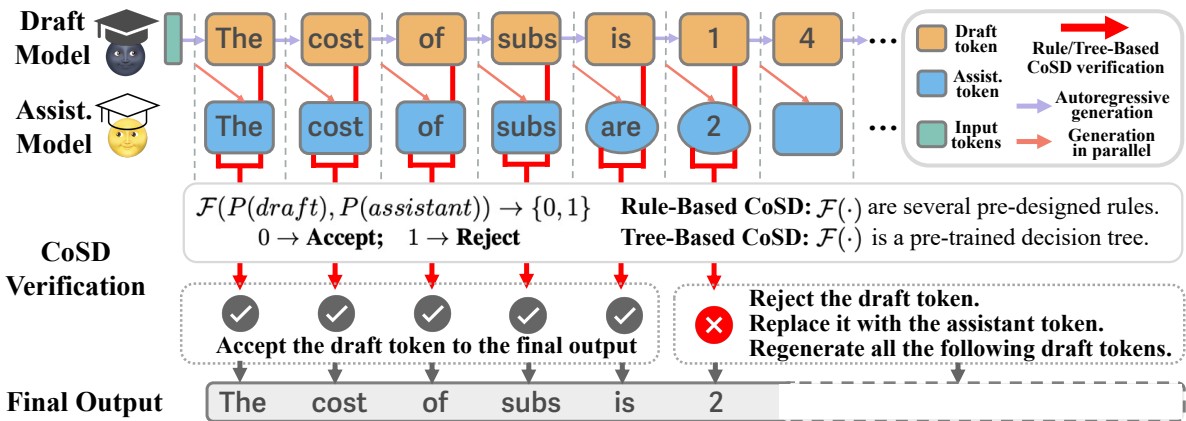

*Figure 1.* The workflow of collaborative speculative decoding.

in following sections.

### 3.1. Generation.

The generation process follows the principles of Speculative Decoding. First, the draft model $\mathcal{M}_p$ generates a sequence of tokens autoregressively:

$$\text{for } i = 1 \text{ to } K \text{ do}$$
$$\widetilde{x}_{t+i} \sim \mathcal{M}_p(x|x_1, \ldots, \widetilde{x}_{t+i-1}), \tag{1}$$

Here, $\widetilde{x}_{t+i}$ represents the token predicted by the draft model at position $i$, selected as the token with the highest probability. The sequence $\widetilde{x}_{t+1}, \ldots, \widetilde{x}_{t+K}$ is generated autoregressively and produced sequentially.

After the draft sequence is generated, the assistant model $\mathcal{M}_q$ is used to verify these tokens. The assistant model generates tokens in parallel:

$$i = 1, \ldots, K \text{ in parallel do}$$
$$\hat{x}_{t+i} \sim \mathcal{M}_q(x|x_1, \ldots, \widetilde{x}_{t+i-1}), \tag{2}$$

Note that we already have all the draft tokens $\widetilde{x}_{t+1}, \ldots, \widetilde{x}_{t+K}$ when we generate the assistant tokens. Thus, all the $\hat{x}_{t+i}$ in Eq. (2) can be generated in parallel. The process can also handle cases where the draft and assistant models use different tokenizers. In such cases, the draft sequence is first decoded by the draft model's tokenizer and then encoded by the assistant model's tokenizer:

$$i = 1, \ldots, K \text{ in parallel do}$$
$$[x_1, \ldots, \widetilde{x}_{t+i-1}] \xrightarrow[T_p]{decode} \text{Texts} \xrightarrow[T_q]{encode} [x_1^*, \ldots, x_n^*],$$
$$\hat{x}_{t+i} \sim \mathcal{M}_q(x|x_1^*, \ldots, x_n^*), \tag{3}$$

where $T_p$ and $T_q$ are the tokenizers of the draft model and the assistant model respectively. The draft sequence is first decoded into texts by $T_p$ and then encoded by $T_q$ to fit the assistant model.

### 3.2. Verification

After the generation, we have a draft sequence $\widetilde{x}_{t+1}, \ldots, \widetilde{x}_{t+K}$ and an assistant sequence $\hat{x}_{t+1}, \ldots, \hat{x}_{t+K}$, along with the corresponding probabilities $\mathcal{M}_p(\widetilde{x}_{t+i})$ and $\mathcal{M}_q(\hat{x}_{t+i})$. We then use this information to verify whether to keep the draft token $\widetilde{x}_i$ or replace it with the assistant token $\hat{x}_i$ and thus ensemble the model knowledge. In order to make CoSD suitable for a wider range of tasks, we propose two strategies for verification. The first strategy, called Rule-Based Verification, applies clear rules to decide whether to select the draft token or the assistant token. The second strategy, i.e., Tree-Based Verification, involves training a decision tree to classify and select between the draft and assistant tokens.

**Rule-Based Verification.** In Rule-Based Verification, the system applies simple yet general rules to determine whether the draft token $\widetilde{x}_{t+i}$ should be replaced by the assistant token $\hat{x}_{t+i}$. The intuition behind these rules is that if the draft model predicts a token with low confidence and the assistant model offers a higher-confidence alternative, the draft token should be replaced. The following rules define the verification process:

$$\widetilde{x}_{t+i} \neq \hat{x}_{t+i}, \tag{4}$$
$$\mathcal{M}_p(\widetilde{x}_{t+i}) < \alpha, \tag{5}$$
$$\mathcal{M}_q(\hat{x}_{t+i}) > \beta \cdot \mathcal{M}_p(\widetilde{x}_{t+i}), \tag{6}$$

These conditions check whether (1) the draft and assistant tokens differ, (2) the draft token has a probability below a threshold $\alpha$, and (3) the assistant token has a probability sufficiently higher than the draft token's probability by a factor of $\beta$. If all conditions are met, the draft token is replaced with the assistant token.

Intuitively, the Rule-Based Verification can be explained as follows: if the draft model is uncertain and the assistant model provides a better alternative, the system opts for

---

**Algorithm 1** Workflow of CoSD

---

**Input:** Draft model $\mathcal{M}_p$, assistant model $\mathcal{M}_q$, input sequence $x_1, \ldots, x_t$, predefined hyperparameters $\alpha, \beta$ and trained decision tree $\mathcal{T}$;

**Output:** Output sequence $x_{t+1}, \ldots, x_{t+K}$;

**Generation**
1: **for** $i$ in $0, 1, \ldots, K$ **do**
2:     $\widetilde{x}_{t+i} \sim \mathcal{M}_p(x|x_1, \ldots, \widetilde{x}_{t+i-1})$     #**Generate draft in an auto-regressive manner.**
3: **end for**
4: Verify the draft in parallel:
5: $i = 1, \ldots, K$ **in parallel do**
6:     $\hat{x}_{t+i} \sim \mathcal{M}_q(x|x_1, \ldots, \widetilde{x}_{t+i-1})$,     #**Generate the assistant sequence in parallel.**
7: Send both $\widetilde{x}_1, \ldots, \widetilde{x}_K, \hat{x}_1, \ldots, \hat{x}_K$, and all related probabilities $\mathcal{M}_p(\widetilde{x}_i), \mathcal{M}_q(\hat{x}_i)$ to verification.

**Verification**
8: **for** $i$ in $0, 1, \ldots, K$ **do**
9:     **if** $\widetilde{x}_{t+i} \neq \hat{x}_{t+i}$ **and** $\mathcal{M}_p(\widetilde{x}_{t+i}) < \alpha$ **and** $\mathcal{M}_q(\hat{x}_{t+i}) > \beta \cdot \mathcal{M}_p(\widetilde{x}_{t+i})$ **then**        **or**
10:     **if** $\widetilde{x}_{t+i} \neq \hat{x}_{t+i}$ **and** $\mathcal{T}(\mathcal{M}_p(\widetilde{x}_{t+i}), \mathcal{M}_q(\hat{x}_{t+i})) = 1$ **then**
11:        $x_{t+i} \leftarrow \hat{x}_{t+i}$
12:        $t \leftarrow t + i$
13:        **Exit loop, go to Generation**
14: **end for**

---

the assistant's prediction. If a replacement is made, the sequence is updated, and the draft model regenerates from that point onward.

**Tree-Based Verification.** For domain-specific applications, Rule-Based Verification may not always be optimal. It is necessary to improve performance in specialized domains, such as healthcare (Poonia & Al-Alshaikh, 2024), smart home (Amru et al., 2024), or math (Mazraeh et al., 2024). Therefore, we design the Tree-Based Verification method, which involves training a decision tree to decide when to replace a draft token with an assistant token. Training the decision tree on specific domain data allows for a more accurate assessment of knowledge fusion performance within those particular contexts. Specifically, our decision tree $\mathcal{T}$ takes two probabilities, $\mathcal{M}_p(\widetilde{x}_{t+i})$ and $\mathcal{M}_q(\hat{x}_{t+i})$, as inputs. The decision tree's output $\mathcal{T}(\mathcal{M}_p(\widetilde{x}_{t+i}), \mathcal{M}_q(\hat{x}_{t+i})) \in \{0, 1\}$ indicates whether to use the draft token ($y_i = 0$) or replace it with the assistant token ($y_i = 1$).

To train a decision tree suitable for specific domains, we first select a commonly used benchmark dataset $D$ for this domain (e.g., GSM8K (Cobbe et al., 2021) in math) with several input and ground-truth output pairs, *i.e.*, $x_1, \ldots, x_t$ and $x_{t+1}, \ldots, x_{t+K}$. We iterate through all the tokens in the ground-truth output in each pair. For the $i$-th token, we concatenate the input sequence and the first $i - 1$ tokens of output sequences. Then, we feed the concatenated input $x_1, \ldots, x_{t+i-1}$ into the two models separately to obtain the predicted next token $\widetilde{x}_{t+i}$, $\hat{x}_{t+i}$ and their corresponding probabilities $\mathcal{M}_p(\widetilde{x}_{t+i})$, $\mathcal{M}_q(\hat{x}_{t+i})$. This probability pair

is one training sample of the decision tree. As for the related ground-truth label, we have three rules:

- If $\widetilde{x}_{t+i} = x_{t+i}$, we assign the label $y_i = 0$ to encourage the decision tree to select the draft token.
- If $\widetilde{x}_{t+i} \neq x_{t+i}$ and $\hat{x}_{t+i} = x_{t+i}$, we assign the label $y_i = 1$ to encourage the decision tree to select the assistant token.
- If neither $\widetilde{x}_{t+i}$ nor $\hat{x}_{t+i}$ match the target, we drop the sample and continue the loop with $i \leftarrow i + 1$.

We iterate through all the input-output pairs and finally construct the training data sample in the form of $\{[\mathcal{M}_p(\widetilde{x}_i), \mathcal{M}_q(\hat{x}_i)], y_i\}$. In the training process, we aim to train the decision tree classifier $\mathcal{T} : \mathbb{R}^2 \to \{0, 1\}$ to minimize the difference between the predicted label and the ground truth:

$$\min_{\mathcal{T}} \sum_{i=1}^{N} [y_i \log(\mathcal{T}(\mathcal{M}_p(\widetilde{x}_i), \mathcal{M}_q(\hat{x}_i))) \\ + (1 - y_i) \log(1 - \mathcal{T}(\mathcal{M}_p(\widetilde{x}_i), \mathcal{M}_q(\hat{x}_i)))]. \tag{7}$$

After training, our decision tree can predict whether to choose the draft token or the assistant token based on the two input probabilities. If the decision tree predicts 1, the same as the rule-based verification, we replace the token, update the accepted token number, and send the new input sequence back to the generation. Since the decision tree is trained on a dataset specific to the corresponding domain, using this decision tree to fuse the model outputs can achieve better results in that domain.

*Table 1.* LLM pairs in the experiments.

| Methods | Draft Model | Assist. Model | Simulated Scenario |
|---------|-------------|---------------|---------------------|
| **Pair 1** | Mistral 7B DARE | Mistral 7B Mixed | Complementary Knowledge Fusion |
| **Pair 2** | Llama 3 Wissenschaft 8B | Llama 3 Bophades 8B | Complementary Knowledge Fusion |
| **Pair 3** | Mistral 7B (Jiang et al., 2023) | Mistral Math 7B | Catastrophic Forgetting Recovery |
| **Pair 4** | TinyLlama (Zhang et al., 2024) | Llama 2 Chat | Capacity Imbalance |
| **Pair 5** | Llama 2 Chat (Touvron et al., 2023) | WizardMath (Luo et al., 2023) | Different Tokenizers |
| **Pair 6** | Llama 2 Chat | DeepSeek Coder (Guo et al., 2024) | Different Tokenizers |

## 4. Experiment

### 4.1. Experimental Settings

**Scenarios, Models, and Benchmarks.** We evaluate CoSD and compare it against several baselines in scenarios that reflect common use cases where users may seek to fuse the knowledge of multiple LLMs. These scenarios include: (i) **Complementary Knowledge Fusion:** The fused LLMs have complementary knowledge, and users hope that the knowledge fusion system can perform as well as the best model for each task across all tasks; (ii) **Catastrophic Forgetting Recovery:** The fused models are one base model and a model fine-tuned from the base model. Fine-tuning improves performance in certain domains but reduces the performance in other domains due to catastrophic forgetting. Users expect to heal the catastrophic forgetting by fusing the knowledge of the two LLMs; (iii) **Capacity Imbalance:** Users use a small draft model and adopt an API of the assistant model with a much larger capacity. The fusion system is expected to perform similarly to the assistant model; (iv) **Different Tokenizers:** Fuses the LLMs with different tokenizers. To simulate these scenarios, we carefully selected six pairs of LLMs from the HuggingFace repository (Jain, 2022), representing each of the four use cases outlined above. Table 1 lists the model pairs and the corresponding simulated scenarios.

For all the scenarios and model pairs, we use MMLU (Hendrycks et al., 2020), GSM8K (Cobbe et al., 2021), HumanEval (Chen et al., 2021), Hellaswag (Zellers et al., 2019), and TruthfulQA (Lin et al., 2021) as the evaluation benchmark. We use tinyBenchmarks (Polo et al., 2024) for all the benchmarks except HumanEval to further increase the efficiency of experiments. These benchmarks test general question-answering, mathematical reasoning, and coding capabilities, providing a comprehensive assessment of the models' abilities across different domains. By using these benchmarks, we can evaluate the effectiveness of CoSD and the baselines in fusing complementary knowledge across diverse tasks and model configurations.

**Baselines.** We use tree baselines in the experiment. **Speculative Decoding:** It also uses a draft model and an assistant model to generate the output. However, it adopts a different verification algorithm that replaces the draft token when

$\frac{\mathcal{M}_p(\widetilde{x}_i)}{\mathcal{M}_q(\hat{x}_i)} < U(0,1)$. **Average Decoding:** It averages the predicted probabilities of the draft model and the assistant model and chooses the final output from the averaged probabilities. **Co-LLM (Shen et al., 2024):** It trains a single layer to classify the hidden state of a base model. The output probability of the layer decides to use the base model generation or evoke an assistant model to help generation.

**Hyperparameters.** We run CoSD with the following settings. For Rule-Based CoSD, we set $\alpha = 0.5$ and $\beta = 0.5$, which were determined to be the optimal and most transferable parameters based on our analysis in Figure 2. For Tree-Based CoSD, we randomly select three samples from the AlpacaEval dataset to train the decision tree. It is important to note that we use MMLU, GSM8K, and HumanEval as our benchmarks. Consequently, the training data for the decision tree do not overlap with the test data, creating a more realistic scenario to evaluate the decision tree's transferability across different tasks and domains.

### 4.2. Experimental Results

**Fusing LLMs with Complementary Domain Knowledge.** We first evaluated the performance of different methods for fusing LLMs with complementary knowledge, with results shown in the pair 1 and pair 2 columns of Table 2. Both CoSD-Rule and CoSD-Tree consistently outperformed the baseline methods in terms of averaging performance. For instance, in pair 1, CoSD-Rule and CoSD-Tree achieved averaged scores of 52.41 and 51.24 on, surpassing all the baselines and both the draft model and the assistant model. Besides, CoSD-Rule also achieves the best performance on MMLU, HumanEval, and Hellaswag. In pair 2, CoSD matches the performance of the better model for each task across all tasks. For example, CoSD achieves a similar MMLU performance to the draft model and a similar performance on GSM8K and HumanEval to the assistant model in pair 2. Compared with CoSD, Speculative Decoding only performs similarly to the assistant model, thus will be more suitable to the scenario when the assistant model is much stronger than the draft model. Average Decoding fuses model knowledge but only achieves an average accuracy across tasks, unlike CoSD, which integrates the strengths of different LLMs. Co-LLM's performance is the closest to

*Table 2.* The results of fusing LLMs with complementary knowledge and the same tokenizer. Pair 1 and pair 2 are complementary knowledge fusion results. Pair 3 simulates a catastrophic forgetting healing scenario, and pair 4 is a disparate capacity LLM fusion result.

| Models | Benchmarks | Draft | Assist. | Spec. Decoding | Avg. Decoding | Co-LLM | CoSD-Rule | CoSD-Tree |
|---|---|---|---|---|---|---|---|---|
| **Pair 1** | **MMLU** | 65.82 | 59.26 | 59.33 | 62.22 | 60.40 | **65.06** | 63.71 |
| | **GSM8K** | 31.20 | 42.19 | 33.36 | 38.33 | **38.85** | 36.81 | 37.24 |
| | **HumanEval** | 28.66 | 31.10 | 14.02 | 25.60 | 29.91 | **31.34** | 28.29 |
| | **Hellaswag** | 84.35 | 79.71 | 80.25 | 81.44 | 82.58 | **85.19** | 83.17 |
| | **TruthfulQA** | 43.92 | 40.62 | **44.18** | 40.52 | 42.78 | 43.65 | 43.77 |
| | **AVG.** | 50.79 | 50.58 | 46.23 | 49.62 | 50.90 | **52.41** | 51.24 |
| **Pair 2** | **MMLU** | 54.81 | 52.02 | 53.20 | 52.31 | 55.25 | 56.97 | **58.37** |
| | **GSM8K** | 39.79 | 51.02 | 43.85 | 43.89 | 41.04 | **45.72** | 41.89 |
| | **HumanEval** | 21.34 | 43.90 | 39.02 | 38.41 | 37.25 | **39.10** | 36.22 |
| | **Hellaswag** | 86.41 | 82.36 | 82.52 | 86.39 | 85.64 | **86.96** | 86.84 |
| | **TruthfulQA** | 56.90 | 49.54 | 51.29 | 54.45 | 55.07 | 55.09 | **55.40** |
| | **AVG.** | 47.85 | 55.77 | 53.98 | 55.09 | 54.85 | **56.77** | 55.74 |
| **Pair 3** | **MMLU** | 61.45 | 46.59 | 43.39 | 56.60 | 58.78 | 62.41 | **63.87** |
| | **GSM8K** | 25.01 | 35.43 | 33.10 | 36.61 | 37.15 | **45.47** | 33.85 |
| | **HumanEval** | 27.44 | 9.76 | 10.97 | 18.90 | 21.88 | **25.61** | 23.17 |
| | **Hellaswag** | 82.71 | 73.86 | 80.67 | 75.73 | 79.03 | **82.81** | 81.95 |
| | **TruthfulQA** | 35.45 | 48.50 | 36.23 | 41.88 | 44.60 | 44.49 | **46.01** |
| | **AVG.** | 46.37 | 42.83 | 40.87 | 45.94 | 48.29 | **52.16** | 49.77 |
| **Pair 4** | **MMLU** | 32.13 | 47.65 | 47.30 | 42.62 | 47.47 | 47.84 | **48.15** |
| | **GSM8K** | 3.36 | 15.63 | **14.63** | 12.12 | 11.97 | 12.52 | 12.29 |
| | **HumanEval** | 8.53 | 12.20 | 10.39 | 12.55 | 11.73 | **12.80** | 10.54 |
| | **Hellaswag** | 60.76 | 77.95 | 75.94 | 68.77 | 73.19 | 73.23 | **74.08** |
| | **TruthfulQA** | 30.47 | 32.09 | 29.98 | 32.35 | 30.64 | 32.17 | **32.58** |
| | **AVG.** | 27.05 | 37.10 | 35.65 | 33.68 | 35.00 | **35.71** | 35.53 |

*Table 3.* Fusing LLMs with different tokenizers.

| Models | Benchmarks | Draft | Assist. | Char-ED | CoSD-Rule | CoSD-Tree |
|---|---|---|---|---|---|---|
| **Pair 5** | **MMLU** | 47.65 | 40.61 | 44.29 | 50.65 | **52.13** |
| | **GSM8K** | 15.63 | 51.13 | 37.54 | **44.88** | 37.01 |
| **Pair 6** | **MMLU** | 47.65 | 59.63 | 52.51 | **57.33** | 55.20 |
| | **HumanEval** | 8.53 | 73.17 | 59.04 | **59.88** | 51.42 |

CoSD, but since it requires training on specific datasets, its transferability across different datasets is inferior to CoSD.

These results highlight the effectiveness of CoSD, particularly the superior fusion capabilities across multiple benchmarks and model pairs. With the support of experimental results, we state that if users have multiple language models with different specialties but are unsure when to use each model, CoSD always provides better overall performance.

**Catastrophic Forgetting Recovery.** We select a Mistral base model and a fine-tuned math Mistral model for pair 3 in Table 2 to simulate the catastrophic forgetting recovery. We found that CoSD-Rule performs particularly well on this type of task. It not only recovers from forgetting across all benchmarks but also outperforms both the draft and assis-

tant models on MMLU, GSM8K, and Hellaswag. These results suggest that CoSD can further enhance the original performance of both models by enabling collaboration between them.

**Fusing LLMs with disparate capacity.** When the assistant model has a much larger capacity than the draft model, the model fusion system is supposed to achieve a similar performance to the draft model. Speculative Decoding is more suited for this task because its verification strategy tends to replace more draft tokens with assistant tokens. However, CoSD results in pair 4, Table 2 are still comparable to Speculative Decoding. For instance, CoSD-Rule has higher HumanEval scores than Speculative Decoding, while CoSD-Tree has higher MMLU, Hellaswag, and TruthfulQA

scores than Speculative Decoding. These results on LLMs with disparate capacities indicate that CoSD is not only applicable to complementary knowledge LLM fusion but also to efficient inference tasks. When the draft model is smaller and the assistant model is larger, our CoSD can achieve performance similar to the assistant model. At the same time, since the assistant model only performs parallel verification, CoSD still has more efficient inference compared to using the assistant model alone.

**Fusing LLMs with Different Tokenizers.** Although CoSD needs to decode and then encode the sequences during the verification when the models have different tokenizers, which sacrifices some efficiency, it can still effectively fuse the model knowledge. In the experiments, we fuse a Llama 2 Chat and a WizardMath to evaluate the CoSD performance on MMLU and GSM8K. We fuse a Llama 2 Chat and a Deepseek Coder to evaluate CoSD on MMLU and HumanEval. Results are shown in Table 3. CoSD outperforms the character-wise averaging method CharED (Gu et al., 2024) in both model pairs and benchmarks. We do not include other baselines since they are not applicable to the different tokenizer settings.

**Ablation Studies.** We have several tunable hyperparameters in CoSD. In Rule-Based CoSD, we have $\alpha$ and $\beta$ that determine the rules to replace the draft tokens. In Tree-Based CoSD, the training data and hyperparameters influence the performance of the decision tree. Thus, we use ablation experiments to identify the impact of these hyperparameters on the final model performance, allowing us to determine the optimal and transferable hyperparameter settings. Figure 2 shows the relationship between $\alpha$, $\beta$ values in Rule-Based CoSD and model performance. The x-axis represents the values of $\alpha$, and the y-axis represents the values of $\beta$. The numbers in the small squares represent the sum score of MMLU and GSM8K, which reflect the overall model performance of CoSD. We can see that with $\alpha = 0.5, 0.75$ and $\beta = 0.5, 0.75$, Rule-Based CoSD perform consistently well in the two model pairs. We ultimately selected $\alpha = 0.5, \beta = 0.5$ as the general hyperparameters in our experiments. We believe this setting effectively integrates the knowledge of the models.

Table 5 displays the impact of the tree training dataset on Tree-Based CoSD. The decision tree trained on different datasets performs relatively consistently, even when the training set is not in the same distribution with any benchmark (e.g., AlpacaEval achieves good results across all three benchmarks). When the decision tree's training set shares the same distribution as a particular benchmark, CoSD-Tree tends to perform slightly better on that benchmark. Therefore, if users are aware of the model's application scenario, they can use the corresponding benchmark from that task

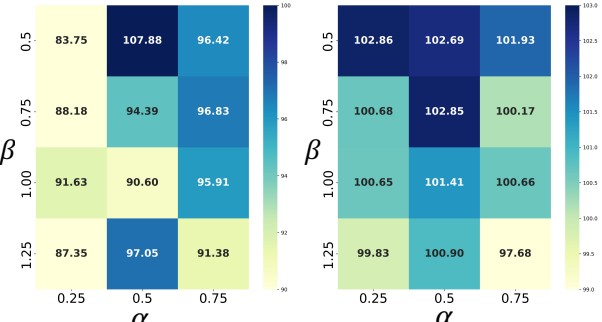

*Figure 2.* The sum score of MMLU and GSM8K with various $\alpha$, $\beta$ settings on pair 1 (left figure) and pair 2 (right figure).

to train the decision tree. This would result in a domain-adapted tree that is better suited to the specific task. In addition, as mentioned in the table title, we use very few samples to train the decision tree, thus training decision trees introduces almost no additional computational overhead.

**Case Studies.** We use an example in GSM8K to demonstrate how CoSD effectively combines the knowledge of two models in Table 4. CoSD replaces the red tokens generated by the draft model with the green tokens from the assistant model. We display the answer from the single draft model and the assistant model in Table 10 in the appendix. Neither the draft model nor the assistant generates the correct result when used alone. The main issue with the draft model is its weak mathematical calculation ability (*e.g.,* in the fourth line, it calculates the tax as 20% of 20 to be 10, instead of the correct answer 4). On the other hand, the assistant model performs well in terms of mathematical calculations but lacks the logical rigor of the draft model. It fails to compute the subtotal without the tip first, leading to the incorrect final calculation. CoSD effectively integrates the strengths of both models. For instance, in CoSD-Rule, in the fifth line, the assistant model rejects the draft model's incorrect computation of 20% of 20 = 10 and instead uses the correct calculation of 20 * 0.2 = 4, successfully avoiding the error in the draft model's tax calculation. In the sixth line, the draft model correctly leads to generate the subtotal of $24, so in the final step, CoSD-Rule computes the simpler 24 + 5 instead of the more complicated 15 + 3 + 2 + 5, resulting in the correct answer.

Also, there are situations that CoSD makes wrong decisions. As shown in Table 9 in Appendix B, CoSD does not always select the correct answer. In the above example, the draft model made the correct choice with high confidence, so the final generation retained the correct answer. However, in the example below, while the draft model also made the correct choice, the assistant model provided an incorrect answer with higher confidence, leading to the final output being changed to the wrong answer. This demonstrates that using confidence as the criterion does not guarantee selecting the

*Table 4.* An example of how COSD polish the draft generation in GSM8K dataset. The table shows the outputs for a GSM8K question generated by CoSD-Rule and CoSD-Tree. Output tokens that are not highlighted represent accepted draft tokens, while tokens marked in pink are rejected draft tokens, followed by the assistant tokens that replace the rejected ones highlighted in green.

| Model | Example |
|---|---|
| **Question** | Rory orders 2 subs for \$7.50 each, 2 bags of chips for \$1.50 each and 2 cookies for \$1.00 each for delivery. There's a 20% delivery fee added at check out and she wants to add a \$5.00 tip. What will her delivery order cost? Correct answer: 29 |
| **CoSD Rule** | The cost of the subs is 2 ~~*~~ x \$7.50 = ≪2*7.50=15≫15 
 The cost of the bags of chips is 2 x \$1.50 = ≪2*1.50=3≫3 
 ~~We~~ The ~~delivery~~ total ~~final~~ sub ~~is~~ before ~~tax~~ the ~~tax~~ delivery fee is 
 ~~15~~ \$ 15 + \$3 + \$2 = \$≪15+3+2=20≫20 
 The delivery fee is 20% of \$20 ~~=~~ , so the delivery fee is \$20 x 0.20 = ≪20*0.20=4≫4 
 The total ~~cost~~ before the tip is \$20 + \$4 = \$≪20+4=24≫24 The tip is \$5.00 
 The total ~~of~~ delivery ~~is~~ order ~~will~~ cost is \$24 + \$5 = \$≪24+5=29≫29 |
| **CoSD Tree** | The cost of the subs is 2 ~~*~~ x \$7.50 = ≪2*7.5=15≫15 
 The cost of the ~~bags~~ chips is 2 x \$1.50 = ≪2*1.5=3≫3 
 The cost of the cookies: 2 x \$1.00 = ≪2*1=2≫2 
 The ~~cost~~ subtotal ~~of~~ before the delivery fee: 15 + 3 + 2 = ≪15+3+2=20≫20 
 The 20% delivery fee: 20% of 20 = ≪20 ~~%~~ * ~~2~~ .2=4≫4 
 The total cost of the order before the tip: 20 + 4 = ≪20+4=24≫24 
 The total cost of the order ~~is~~ by adding ~~all~~ the tip: 24 + 5 = ≪24+5=29≫29 |

*Table 5.* Training the decision tree with different datasets in pair 3. Each column represents a decision tree trained by the dataset in the column header. We use 10 samples of MMLU, 3 samples of each other datasets to train the decision tree.

| Benchmarks | MMLU | GSM8K | HumanEv. | AlpacaEv. |
|---|---|---|---|---|
| **MMLU** | **63.94** | 60.88 | 61.23 | 63.87 |
| **GSM8K** | 35.04 | **37.17** | 30.08 | 33.85 |
| **HumanEv.** | **25.62** | 23.04 | 23.09 | 23.17 |

*Table 6.* Efficiency of LLM Knowledge Fusion. Token latency represents the averaged time to generate one token, and acceptance rate refers to the proportion of draft tokens that are not replaced. Typically, the higher the latter, the lower the former, as fewer tokens require regeneration. Experiments are done by pair 3.

| Methods | Token Latency (ms) | Acceptance Rate |
|---|---|---|
| **Spec. Decoding** | 131.22 | 0.89 |
| **CoSD-Rule** | 132.31 | 0.81 |
| **CoSD-Tree** | 135.82 | 0.77 |

correct option but can only aim to choose the correct answer with a higher probability.

**Efficiency.** Since we perform fusion during the inference, efficiency is a major advantage of our approach. We compared the time overhead of COSD with the baselines. We use token latency and acceptance rate as the metrics for efficiency. As displayed in Table 6, Speculative Decoding has the lowest latency among all methods, since it makes the least token replacement. However, although COSD methods replace a few more tokens, the increase in total latency is almost negligible. Considering that COSD has the best knowledge fusion performance, we have achieved a better balance between efficiency and effectiveness.

## 5. Conclusion

In this paper, we fuse the LLM knowledge in a simple yet effective way. Our algorithm COSD takes the probabilities of predicted tokens from two LLMs as the feature to verify whether to keep the draft token or adopt the assistant token. The verification strategy can be either a rule-based or a pre-trained decision tree. Our extensive experiments show that COSD performs better than the state-of-the-art methods across 6 LLM pairs and 5 benchmarks. Compared to previous works, COSD has superior knowledge fusion ability, a broader range of application scenarios, and comparable efficiency. It works well in scenarios including complementary knowledge fusion, catastrophic forgetting recovery, knowledge fusion with disparate model capacity, and knowledge fusion with different tokenizers. COSD makes it possible for ordinary users to fuse the LLM knowledge with only the API queries, without any training or fine-tuning of LLMs, or requirements of white-box LLM information such as hidden states. It provides users with better tools to manipulate LLMs in wider application scenarios.

## Impact Statement

This paper presents work whose goal is to improve the test-time knowledge fusion algorithm for language models. There are many potential societal consequences of our work, none which we feel must be specifically highlighted here.

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

# A. Limitations

While COSD demonstrates strong performance across various scenarios, it is important to acknowledge its limitations. This section highlights cases where COSD may not be applicable and tasks that it fails to address. Identifying these constraints provides clarity on its scope of use and helps guide future improvements. Below, we outline two specific limitations:

(1) When the two collaborating models are of similar size and one significantly outperforms the other, COSD offers no advantage over using only the better model. In this case, using the better model only is sufficient. This also requires the user to have prior knowledge of the performance of the two models on different benchmarks and to determine that one model is significantly better than the other. If the user is uncertain, we still recommend using COSD to ensure the best results.

(2) Another limitation of COSD is that it cannot guarantee the replaced assistant token is always better than the discarded draft one. It relies on the confidence scores of the models, which are not always perfectly aligned with token quality. The algorithm selects the output of the more confident model, aiming to maximize the likelihood of choosing a better token, but this approach may occasionally lead to suboptimal results.

# B. Additional Experiments and Discussion

**The Average Iterations in COSD.** The number of iterations required during collaborative decoding depends on the maximum length of the model output. Table 7 reports the average number of iterations on the GSM8K dataset for different maximum lengths.

| Max Length | CoSD-Rule | CoSD-Tree | Spec. Dec. |
|:---:|:---:|:---:|:---:|
| **128** | 11.41 | 13.58 | 9.77 |
| **256** | 15.29 | 16.01 | 14.20 |
| **512** | 21.23 | 21.95 | 18.51 |

*Table 7.* Average number of iterations for different maximum output lengths.

Although the number of iterations scales with the output length, it does not directly imply a proportional increase in generation time. As the number of accepted tokens grows, the number of tokens requiring regeneration decreases significantly. For instance, with a maximum output length of 128, the average number of iterations is 11, but the total generated output length remains around 300 tokens. This highlights the efficiency of our approach in reducing redundant generation.

**Collaborate with More LLMs.** Our COSD also supports multiple collaborating models. Table 8 presents the results when three models are used for collaboration:

| Dataset | Draft | Assist. 1 | Assist. 2 | CoSD-Rule | CoSD-Tree |
|:---:|:---:|:---:|:---:|:---:|:---:|
| **MMLU** | 32.13 | 47.65 | 35.62 | 44.14 | 46.48 |
| **GSM8K** | 3.36 | 15.63 | 8.33 | 15.85 | 14.02 |

*Table 8.* Performance of three collaborator LLMs.

In this setup, the draft model is TinyLlama, while the assistant models are Llama 2 Chat 7b and Llama-7b. Our findings demonstrate that involving additional models improves prediction accuracy. Table 8 demonstrates that when three models collaborate if one significantly outperforms the other two, the final system will achieve performance close to that of the best model. This indicates that our algorithm is effective when applied to more than two models. With sufficient LLMs, we can also better utilize training data, even when certain samples are excluded.

**The Case Study of MMLU.** While COSD is effective in many cases, there are instances where it makes incorrect decisions, highlighting its limitations. As shown in Table 9, COSD does not always select the correct answer when the draft model and the assistant model disagree. In the first example, the draft model correctly identified the answer with high confidence, which allowed the final output to retain the accurate result. This showcases the potential of COSD to preserve correct answers when confidence aligns with accuracy.

However, in the second example, the draft model once again made the correct prediction, but the assistant model, despite being incorrect, provided an answer with higher confidence. Consequently, the final output was altered to the wrong answer,

| | Example |
|---|---|
| **Question** | The Yang-shao culture gave way to the Lung-Shan sometime after:
A. 6,000 B.P. B. 5,000 B.P. C. 4,000 B.P. D. 3,000 B.P.
Correct answer: B |

| **Answer** | **Draft** | **Assist.** | **CoSD-Rule** | **CoSD-Tree** |
|---|---|---|---|---|
| | B | D (wrong) | B | B |

| | Example |
|---|---|
| **Question** | Rowena can paint a room in $14$ hours, while Ruby can paint it in 6 hours. If Rowena paints for $x$ hours and Ruby paints for $y$ hours, they will finish half of the painting, while if Rowena paints for $y$ hours and Ruby paints for $x$ hours they will paint the whole room. Find the ordered pair $(x, y)$.
A. $(\frac{11}{10}, \frac{11}{10})$ B. $(\frac{231}{20}, \frac{21}{20})$ C. $(\frac{231}{40}, \frac{21}{40})$ D. (1,1)
Correct answer: C |

| **Answer** | **Draft** | **Assist.** | **CoSD-Rule** | **CoSD-Tree** |
|---|---|---|---|---|
| | C | D (wrong) | D (wrong) | D (wrong) |

*Table 9.* Two examples of how COSD modify the generation in MMLU dataset. The example above demonstrates how COSD helps improve generation quality, while the example below shows instances where COSD sometimes selects incorrect answers.

overriding the draft model's correct prediction. This illustrates a shortcoming of the COSD approach: relying solely on confidence scores as the decision-making criterion does not guarantee correctness. Confidence may reflect certainty but not necessarily accuracy, leading to situations where errors from the assistant model dominate the final outcome.

This limitation suggests that while COSD can improve generation quality by prioritizing higher-confidence predictions, it does so with the assumption that confidence correlates with correctness. In practice, this assumption does not always hold, especially when the assistant model is overconfident in its incorrect predictions. To address this, future improvements could explore additional heuristics or cross-validation mechanisms to better balance confidence with accuracy, ensuring that correct answers are more consistently selected.

| **Model** | **Example** |
|---|---|
| **Question** | Rory orders 2 subs for $7.50 each, 2 bags of chips for $1.50 each and 2 cookies for $1.00 each for delivery. There's a 20% delivery fee added at check out and she wants to add a $5.00 tip. What will her delivery order cost? Correct answer: 29 |
| **Draft Model** | The cost of the subs is 2 * $7.50 = ≪2*7.5=15≫15
The cost of the chips is 2 * $1.50 = ≪2*1.5=3≫3
We add the total cost of the food items by $15 + $3 + $2 = $≪15+3+2=20≫20
The cost of the delivery fee is 20% of the total cost of the order,
so 20% of 20 = ≪20% of 20=10≫10
The total cost of the food items and delivery fee is $20 + $10 = $≪20+10=30≫30
The cost of the tip is $5.00
The total cost of the order is 15 + 3 + 2 + 10 + 5 = ≪15+3+2+10+5=45≫45. |
| **Assist. Model** | The cost of the subs is 2 x $7.5 = $≪2*7.5=15≫15
The cost of the chips is 2 x $1.5 = $≪2*1.5=3≫3
The total cost of her order without any fees is $15 + $3 + $2 = $≪15+3+2=20≫20
The delivery fee is 20% of the total order, so $20 * 0.20 = $≪20*0.20=4≫4
The tip is an additional $5
Therefore, her delivery order will cost $15 + $3 + $2 + $4 + $5 = $≪15+3+2+4+5=35≫35. |

*Table 10.* The answers from the single draft model and assistant model with the question in Table 4.

