# OpenReview forum: "Speculate, then Collaborate: Fusing Knowledge of Language Models during Decoding"
_ICML.cc/2025/Conference — ICML 2025 poster_

### Official Review · Reviewer_fAfk · 2025-03-05

**Overall Recommendation:** 4

**Summary:**

This paper tackles the problem of knowledge fusion across language models while aiming to balance effectiveness and efficiency. The paper proposes CoSD that integrates speculative decoding to accelerate generation and employs probability-based classification for token selection. CoSD provides a flexible and adaptive approach to multi-LLM fusion. The empirical results demonstrate its ability to effectively merge knowledge from multiple models. Comparing to baselines, CoSD has advantages on efficiency and fusion performance.

## update after rebuttal
The authors address most of my concerns. So I keep my positive score.

**Claims And Evidence:**

The paper claims that CoSD improves knowledge fusion performance, enhances efficiency through speculative decoding, and provides flexibility by eliminating explicit model selection. Empirical results basically support these claims, showing multi-LLM integration and faster inference.

**Essential References Not Discussed:**

An early work on model knowledge fusion (not LLMs):
Dong, X. L., & Srivastava, D. (2015, May). Knowledge curation and knowledge fusion: challenges, models and applications. In Proceedings of the 2015 acm sigmod international conference on management of data (pp. 2063-2066).

**Experimental Designs Or Analyses:**

The experiment part focuses on setting up several model pairs to mimic the real-world scenarios when we need to fuse LLM knowledge. CoSD is tested on 5 benchmarks. The experiment settings are valid. However, some questions remain in the hyperparameter part. What is the hyperparameter for the decision tree? Do we need to adjust the parameters during the inference stage, and how can we do this?

**Methods And Evaluation Criteria:**

The method involves running a draft model to generate candidate tokens and verifying them with assistant models before finalizing outputs. Evaluation is conducted on standard LLM benchmarks, measuring fusion performance via accuracy and perplexity while assessing efficiency through decoding speed. Comparisons are made against individual LLM baselines and naive ensemble methods.

**Other Comments Or Suggestions:**

I suggest to show more interesting samples like Table 4 in the appendix.

**Other Strengths And Weaknesses:**

N/A.

**Questions For Authors:**

What is the token latency for only use one LLM in Table 6?

It seems that the CoSD does not directly modify the answers in Table 4. So how does CoSD improve the score on GSM8K? More explanations are needed.

**Relation To Broader Scientific Literature:**

The paper is related to the LLM knowledge fusion field and the efficient inference field. It is an application and augmentation of both fields.

**Theoretical Claims:**

N/A

---

> ### Author Rebuttal · Authors · 2025-03-31
>
> We greatly appreciate the reviewer’s recognition of the strengths of our work. Regarding the weaknesses and questions raised, we address all concerns in detail below.
>
> > However, some questions remain in the hyperparameter part. What is the hyperparameter for the decision tree? Do we need to adjust the parameters during the inference stage, and how can we do this?
>
> We perform the decision tree training with max depth=10. The results of using different max depths can be found here:
>
> | Max Depth | MMLU   | GSM8K  | HumanEval |
> |-----------|--------|--------|-----------|
> | 3         | 61.56  | 34.72  | 21.61     |
> | 5         | 63.00  | 36.52  | 22.34     |
> | 10        | 60.88  | 37.17  | 23.04     |
> | 20        | 60.88  | 37.17  | 23.04     |
>
> Since the performance is mainly influenced by the training dataset rather than the hyperparameters of the decision tree, so it is not necessary to repeatedly tune the hyperparameters — setting a reasonable value is sufficient.
>
> >Essential References Not Discussed
>
> We will add the references to the revised paper. Thanks for pointing out valuable papers to cite and discuss.
>
> >I suggest to show more interesting samples like Table 4 in the appendix.
>
> We will add more samples, especially for the coding samples (i.e., HumanEval test samples) to the appendix and further discuss them in the revised paper.
>
> >What is the token latency for only use one LLM in Table 6?
>
> The token latency of using only one LLM will be around 30-50 ms in Table 6. However, since CoSD is designed to combine complementary knowledge from two models, rather than mimicking one with the other. Therefore, the appropriate baseline is not a single model, but a naive two-model decoding setup where both models generate token-by-token and decide on the final output via a selection mechanism (e.g., our baseline Avg Decoding). In this case, the token latency will be more than 1.5x times higher than CoSD:
>
> | Method         | CoSD-Rule | CoSD-Tree | Avg Decoding | Co-LLM |
> |----------------|-----------|-----------|---------------|--------|
> | Token-Wise Latency | 132.31    | 135.82    | 212.73        | 254.16 |
>
> >It seems that the CoSD does not directly modify the answers in Table 4. So how does CoSD improve the score on GSM8K? More explanations are needed.
>
> We discuss how CoSD improves the score in the Case Study section. In CoSD-Rule, in the fifth line, the assistant model rejects the draft model’s incorrect computation of 20% of 20 = 10 and instead uses the correct calculation of 20 * 0.2 = 4, successfully avoiding the error in the draft model’s tax calculation. In the sixth line, the draft model correctly leads to generate the subtotal of $24, so in the final step, CoSD-Rule computes the simpler 24 + 5 instead of the more complicated 15 + 3 + 2 + 5, resulting in the correct answer.

---

> > ### Comment · Reviewer_fAfk · 2025-04-05
> >
> > The rebuttal addresses my concerns and I keep my positive score.

---

### Official Review · Reviewer_sKWh · 2025-03-09

**Overall Recommendation:** 3

**Summary:**

This paper introduces Collaborative Speculative Decoding (CoSD), a new inference-time algorithm designed to fuse complementary knowledge from multiple LLMs without additional model training or fine-tuning. CoSD leverages a draft model to autoregressively generate initial tokens, which an assistant model then verifies in parallel. A simple, interpretable rule-based or decision-tree-based strategy then determines whether draft tokens should be retained or replaced by tokens proposed by the assistant model, thus effectively integrating diverse knowledge sources.

**Claims And Evidence:**

NA

**Essential References Not Discussed:**

I'm not familiar with related works in this field.

**Experimental Designs Or Analyses:**

The selection of diverse models (e.g., Mistral, Llama, TinyLlama, WizardMath, DeepSeek Coder) effectively demonstrates the generalizability and practical applicability of the proposed method. The authors also used several popular benchmarks such as MMLU, GSM8K, HumanEval, Hellaswag, and TruthfulQA, ensuring comprehensive evaluation.

**Methods And Evaluation Criteria:**

NA

**Other Comments Or Suggestions:**

1. It would be valuable to include a detailed error analysis to identify exactly when and why the decision rules or decision tree sometimes select incorrect tokens.
2. It would be better to analyze the complexity or depth of the decision tree to achieve optimal performance.

**Other Strengths And Weaknesses:**

**Strengths**

1.  This paper clearly defines scenarios where its method performs well and where it may struggle.
2.  Transparent and interpretable methods (rule-based and tree-based verification) enhance the trustworthiness in practice.

**Weaknesses**

1. The training set for the decision tree method is extremely limited (e.g., only 3 samples in some scenarios), raising concerns about robustness and generalization.
2. This paper lacks in-depth statistical significance testing.

**Questions For Authors:**

1. Can you clarify why you chose only three samples for training the decision tree? How sensitive is CoSD's performance to increasing the number or diversity of training samples?
2. Have you considered incorporating more advanced uncertainty estimation methods or ensemble learning approaches (such as Bayesian ensemble methods)? Would this significantly alter CoSD's complexity or performance?
3. Could you provide additional details or analyses on cases where the confidence-based verification fails? What percentage of incorrect token replacements are caused by the assistant model's overconfidence?

**Relation To Broader Scientific Literature:**

Extend the speculative decoding research.

**Theoretical Claims:**

This paper does not provide theoretical justification or analysis to support the claim that training the decision tree with very limited data (eg: only three samples) is sufficient for effective generalization across different tasks or domain. Additionally, while the proposed rule-based mechanism is intuitive, this paper does not include adequate theoretical or statistical analyses examining how well model confidence aligns with correctness in varying scenarios.

---

> ### Author Rebuttal · Authors · 2025-03-31
>
> We greatly appreciate the reviewer’s recognition of the strengths of our work. Regarding the weaknesses and questions raised, we address all concerns in detail below.
>
> > The number of samples for the decision tree. The hyperparameters of the decision tree.
>
> We would like to clarify that using 3 dataset samples (e.g., 3 samples from GSM8K) does not mean the decision tree is trained on only 3 tokens. Each GSM8K sample typically contains more than 50 tokens and can have up to hundreds of tokens. Therefore, 3 samples from GSM8K can provide over 150 training points for the decision tree, which is sufficient given the data sample size (1×2). The sensitivity of CoSD-Tree to the training dataset is reported in Table 5. The relationship between the number of training samples and CoSD-Tree performance is shown below:
>
> | # of Samples | MMLU   | GSM8K  | HumanEval |
> |-----------------------|--------|--------|-----------|
> | 1                     | 62.68  | 35.84  | 22.19     |
> | 3                     | 60.88  | 37.17  | 23.04     |
> | 5                     | 62.34  | 37.62  | 20.94     |
> | 10                    | 61.01  | 36.46  | 20.04     |
>
> We also conducted experiments varying the max depth of the decision tree:
>
> | Max Depth | MMLU   | GSM8K  | HumanEval |
> |-----------|--------|--------|-----------|
> | 3         | 61.56  | 34.72  | 21.61     |
> | 5         | 63.00  | 36.52  | 22.34     |
> | 10        | 60.88  | 37.17  | 23.04     |
> | 20        | 60.88  | 37.17  | 23.04     |
>
> (10 and 20 get the same decision tree)
>
> These results show that CoSD-Tree is not sensitive to the number of training samples or the tree’s hyperparameters. However, it is more sensitive to the type of dataset, as illustrated in Table 5.
>
> > Statistical significance testing
>
> We will report mean ± standard deviation over 5 runs in our experiments in the revised paper.
>
> > It would be valuable to include a detailed error analysis to identify exactly when and why the decision rules or decision tree sometimes select incorrect tokens.
>
> We include illustrative examples in Table 4 and Table 9. For MMLU samples, where a single token determines whether an answer is correct, the decision rule or tree may choose incorrect tokens when the model assigns higher confidence to an incorrect answer. This typically occurs due to hallucination by the assistant model.
>
> For benchmarks requiring chain-of-thought outputs (e.g., GSM8K in Table 4), the replaced tokens are often not directly tied to the final answer (e.g., replacing “we” with “the” in CoSD-Rule). Thus, it can be challenging to pinpoint exactly when and why the decision rule/tree fails. A more informative analysis is to compare CoSD outputs with the outputs of the individual draft and assistant models.
>
> As shown in Table 4, in the fifth line, the assistant model corrects a draft model’s incorrect calculation of 20% of 20 = 10 by replacing it with the correct calculation 20 * 0.2 = 4, avoiding an error in tax computation. In the sixth line, the draft model correctly computes the subtotal $24, leading CoSD-Rule to choose the simpler 24 + 5 over the more verbose 15 + 3 + 2 + 5, producing the correct final answer.
>
> For the tinyMMLU dataset, our experiments show that the wrong replacement rate is around 2%–3% across different model pairs. For Capacity Imbalance tasks (i.e., pair 4), this rate is 0%.
>
> > Have you considered incorporating more advanced uncertainty estimation methods or ensemble learning approaches (such as Bayesian ensemble methods)? Would this significantly alter CoSD’s complexity or performance?
>
> Thank you for the thoughtful suggestion.
>
> We agree that advanced uncertainty estimation techniques and ensemble learning methods, such as Bayesian ensemble models, are valuable and may be worth exploring in future work. However, in this paper, our primary goal is to develop a simple, lightweight, and efficient collaborative decoding framework. To this end, we intentionally adopt minimal decision strategies — a rule-based filter and a shallow decision tree — which are fast, interpretable, and require minimal supervision.
>
> While more advanced methods like Bayesian ensembles may provide stronger theoretical guarantees, we believe they are unlikely to yield significant performance improvements over CoSD in practice, especially considering the strong results we already observe across benchmarks. Moreover, such methods typically involve additional complexity, including performing posterior inference, and evaluating model-specific performance. These extra steps would compromise one of CoSD’s key advantages — being plug-and-play, without requiring per-model evaluation on multiple benchmarks.
>
> We will clarify this design choice and trade-off more explicitly in the revised version. We sincerely thank the reviewer for bringing up this important point.

---

### Official Review · Reviewer_VZYJ · 2025-03-09

**Overall Recommendation:** 4

**Summary:**

This paper addresses the challenge of language model knowledge fusion, aiming to effectively integrate complementary knowledge from multiple LLMs while maintaining efficiency. The authors propose CoSD, a method that classifies output tokens based on their probabilities to achieve fusion and leverages a speculative decoding framework to enhance generation speed. Experimental results demonstrate that CoSD successfully merges knowledge from two or three LLMs while maintaining flexibility, eliminating the need for explicit model selection. The approach is straightforward and empirically effective, offering a practical solution for multi-LLM integration. A deeper analysis of its impact on generation quality and potential trade-offs in diverse scenarios would further strengthen the contribution.

**Claims And Evidence:**

The claims about the paper’s main contribution include the knowledge fusing performance and the effficiency. The paper provides algorithm and evaluations for both points. The evidences are clear to support the claims.

**Essential References Not Discussed:**

Some new papers in this field:

[1]Wan, F., Zhong, L., Yang, Z., Chen, R., & Quan, X. (2024). Fusechat: Knowledge fusion of chat models. arXiv preprint arXiv:2408.07990.

[2]Liu, L., Zhang, D., Li, S., Zhou, G., & Cambria, E. (2024, October). Two Heads are Better than One: Zero-shot Cognitive Reasoning via Multi-LLM Knowledge Fusion. In Proceedings of the 33rd ACM International Conference on Information and Knowledge Management (pp. 1462-1472).

**Experimental Designs Or Analyses:**

I have checked the following experimental parts

(1)The scenarios defined by the paper, including knowledge fusing, catastrophic forgetting healing, capacity imbalance, and different tokenizers. The settings are overall valid, but the knowledge fusing and catastrophic forgetting healing seem similar, might need some explanations.

(2)The picked baselines are valid. The difference between CoSD and baselines is clear.

(3)The hyperparameters seem to be determined by Figure 2. Do other benchmarks follow the same pattern? Need to be explained.

**Methods And Evaluation Criteria:**

This paper evaluate CoSD on 6 LLM pairs and 5 common benchmarks. The experiments use different scenarios such as complementary knowledge fusion and catastrophic forgetting healing. Experiment results delivered the basic ideas. However, the following experiments are not included:

1)What will happen if we swap the draft model and the assistant model? Since the authors claim that users don’t need to choose between LLMs, they may also not want to determine which LLM to be the draft/assistant model. Therefore, the question is will the performance drop significantly if we swap the two models?

2)I’m also very curious about the real samples of the HumanEval dataset. Since the authors already showed the real samples of MMLU (QA) and GSM8K (Math), what will the sample look like in code generation will be interesting. Will the assistant model help repair bugs in the draft generations? A table similar to Table 4 can be helpful.

**Other Comments Or Suggestions:**

If you have any other comments or suggestions (e.g., a list of typos), please write them here.

The algorithm part might be too big (Can it be put in one column?).

The ‘texts’ in Eq. (3) should be inside a bracket. E.g., [x.....x].

**Other Strengths And Weaknesses:**

N/A.

**Questions For Authors:**

If you have any important questions for the authors, please carefully formulate them here. Please reserve your questions for cases where the response would likely change your evaluation of the paper, clarify a point in the paper that you found confusing, or address a critical limitation you identified. Please number your questions so authors can easily refer to them in the response, and explain how possible responses would change your evaluation of the paper.

1)What will happen if we swap the two LLMs?

2)If the two models are in the same size, can CoSD still speed up the inference?

**Relation To Broader Scientific Literature:**

The key contributions are related to the LLM knowledge fusion and the speculative decoding. LLM knowledge fusion aims to fusing the knowledge of different LLMs with complementary knowledge. Speculative decoding focus on speeding up the inference with collaboration between large annd small LMs. CoSD in this paper seems to combine the advantage of the two fileds by specifically designed algorithm.

**Theoretical Claims:**

The paper have no strict theoretical proofs. I have checked the algorithms and the equations and there is no significant flaws.

---

> ### Author Rebuttal · Authors · 2025-03-31
>
> We greatly appreciate the reviewer's recognition of the strengths of our work. Regarding the weaknesses and questions raised, we address all concerns in detail below.
>
> >What will happen if we swap the draft model and the assistant model? Since the authors claim that users don’t need to choose between LLMs, they may also not want to determine which LLM to be the draft/assistant model. Therefore, the question is will the performance drop significantly if we swap the two models? What will happen if we swap the two LLMs?
>
> We performed experiments on swapping the draft model and the assistant model. Here are the results of pair 2:
>
> | Benchmarks | Draft | Assist. | Spc. Dec. | CoLLM | CoSD-R | CoSD-T |
> |------------|-------|---------|-----------|--------|--------|--------|
> | MMLU       | 52.02 | 54.81   | 54.23     | 53.92  | 54.17  | 56.18  |
> | GSM8K      | 51.02 | 39.79   | 41.28     | 45.75  | 49.52  | 48.88  |
> | HumanEval  | 43.90 | 21.34   | 25.17     | 36.90  | 42.31  | 43.62  |
> | **Avg**    | 48.98 | 38.65   | 40.23     | 45.52  | 48.67  | **49.56** |
>
> The results show that CoSD still outperforms all the baselines. We will add these results to the revised paper.
>
> >I’m also very curious about the real samples of the HumanEval dataset. Since the authors already showed the real samples of MMLU (QA) and GSM8K (Math), what the sample will look like in code generation will be interesting. Will the assistant model help repair bugs in the draft generations? A table similar to Table 4 can be helpful.
>
> Thanks for the valuable suggestion. We will add a HumanEval real sample to the revised paper. Our sample shows that the assistant model can modify the draft code to another style (e.g., with more function), which sometimes improves the accuracy and reduces the number of bugs.
>
> >The knowledge fusing and catastrophic forgetting healing seem similar, might need some explanations.
>
> The knowledge fusing task aims to fuse the knowledge of two models with complementary knowledge. The catastrophic forgetting task often fuses the knowledge of one LLM that is only good at one specific task (through fine-tuning) and another model good at all other tasks.
>
> >The hyperparameters seem to be determined by Figure 2. Do other benchmarks follow the same pattern? Need to be explained.
>
> Yes, all the experiments of CoSD-Rule follow the same hyperparameters. We found that the optimal $\alpha$ and $\beta$ values are transferable between models and tasks.
>
> >Comments and Suggestions
>
> Thanks for the suggestions of some details in the paper. We will further polish the paper according to the suggestions.
>
> >If the two models are in the same size, can CoSD still speed up the inference?
>
> Yes, although speculative decoding cannot speed up inference with two models with the same size, our CoSD is used for a different task. CoSD is designed to combine complementary knowledge from two models rather than mimicking one with the other. Therefore, the appropriate baseline is not a single model but a naive two-model decoding setup where both models generate token-by-token and decide on the final output via a selection mechanism (e.g., our baseline Avg Decoding). In this case, the token latency will be more than 1.5x times higher than CoSD:
>
> | Method         | CoSD-Rule | CoSD-Tree | Avg Decoding | Co-LLM |
> |----------------|-----------|-----------|---------------|--------|
> | Token-Wise Latency | 132.31    | 135.82    | 212.73        | 254.16 |
>
> Therefore, CoSD still benefit from the speculative decoding algorithm in this task.

---

> > ### Comment · Reviewer_VZYJ · 2025-04-04
> >
> > I have read the author's rebuttal and the review of other reviewer. I'd love to maintain my accept score.

---

### Official Review · Reviewer_cV7E · 2025-03-17

**Overall Recommendation:** 2

**Summary:**

The paper introduces an algorithm called Collaborative Speculative Decoding (CoSD) that is designed to efficiently fuse the knowledge of multiple Large Language Models (LLMs) together at inference time, without requiring any additional model training. The key idea is to leverage the same inference paradigm followed in standard speculative decoding, except that a general decision rule (instead of the typical rejection sampling) is used to reconcile  differences in token predictions (and as such proceed efficiently in the cases where there is no discrepancy). Experimental results support the effectiveness of this model-merging approach.

**Claims And Evidence:**

The claims are supported. For the most part, the results of CoSD-Tree are somewhat disappointing relative to the naive CoSD-rule approach. However, the results of CoSD-rule are encouraging relative to the evaluated baselines.

**Essential References Not Discussed:**

The related work is adequately discussed.

**Experimental Designs Or Analyses:**

The experiments are sound.

**Methods And Evaluation Criteria:**

Yes, the methods and evaluation criteria make sense. However, I was surprised that no evaluation of speed/efficiency was done, as this seems to be a critical claimed strength over other work that does not take advantage of blockwise parallel decoding. Furthermore, this *should* also be a claim that should be empirically supported, since the typical assumption in speculative decoding is that the drafter model is much more efficient to run sequentially than the verifier model. In this setting, however, both the "drafter" and the "assistant" model are not necessarily computationally that much different, so decoding from each model in parallel (vs. one first, and then the other blockwise parallel) might not have as much benefit?

**Other Comments Or Suggestions:**

- From just looking at Fig. 1 it is unclear what the main differences are between CoSD and standard speculative decoding, apart from generalizing the acceptance / rejection mechanism for draft tokens to incorporate an arbitrary rule or decision tree. I think what you want to emphasize more clearly throughout the paper is that the primary goal is to be able to efficiently merge the predictions of two LLMs that are on equal footing (i.e., vs standard speculative decoding where the verifier is considered to be the better, but slower model). Rather, if I understand correctly, here the speculative decoding aspect is mainly used as a trick to speed up inference and merging of the two models.
- L138: "generated autoregressively and produced sequentially" is redundant.
- The results should have error bars to reflect the effects of randomness in the generations. For example, in theory, Speculative Decoding should have the same average performance as the assistant model, but exhibits a substantial amount of deviation in the table.

**Other Strengths And Weaknesses:**

One weakness which sticks out to me is that this method is unlikely to scale well to multiple collaborating LLMs? With multiple models, the likelihood that a discrepancy arises within just one or a few steps will quickly grow, and speculative decoding will cease to continue to be an efficient strategy (e.g., vs a strategy such as the on in Shen et. al., 2024).

**Questions For Authors:**

The abstract says "CoSD employs a draft model to generate initial sequences and an easy-to-learn rule or decision tree to decide when to invoke an assistant model to improve these drafts." However, from the method description it seems that the assistant model is always "invoked", but just via the speculative decoding procedure. So I am a bit unsure what this is referring to.

**Relation To Broader Scientific Literature:**

The paper introduces a simple method for merging token predictions from different models, that is made efficient by leveraging the same blockwise parallel decoding strategy of speculative decoding. The approach makes sense and is well presented, albeit with fairly limited novelty.

**Theoretical Claims:**

Not applicable.

---

> ### Author Rebuttal · Authors · 2025-03-30
>
> We greatly appreciate the reviewer's recognition of the strengths of our work. Regarding the weaknesses and questions raised, we address all concerns in detail below.
>
> > However, I was surprised that no evaluation of speed/efficiency was done, as this seems to be a critical claimed strength over other work that does not take advantage of blockwise parallel decoding.
>
> We would like to clarify that our method has indeed been evaluated in terms of speed and efficiency. Specifically, **Table 6** in our paper reports latency results, showing that our method achieves nearly the same runtime performance as standard speculative decoding. This confirms that CoSD maintains comparable decoding efficiency. However, we agree that additional efficiency experiments would further highlight our contributions, and we will include them in the revised version of the paper.
>
> > Both the "drafter" and the "assistant" model are not necessarily computationally that much different, so decoding from each model in parallel (vs. one first, and then the other blockwise parallel) might not have as much benefit?
>
> This is an excellent point — In our case, the drafter and assistant models have similar capacities, so such gains might not be expected. However, **our task is fundamentally different**. While speculative decoding aims to approximate the behavior of a large model efficiently, **CoSD is designed to combine complementary knowledge from two models**, rather than mimicking one with the other. Therefore, the appropriate baseline is not a single model, but a **naive two-model decoding** setup where both models generate token-by-token and decide on the final output via a selection mechanism (e.g., our baseline *Avg Decoding*).
>
> For clarity, we provide the comparison below:
>
> | Method    | CoSD-Rule | CoSD-Tree | Avg Decoding | Co-LLM |
> |------|-----|-----|-----|-----|
> | Token-Wise Latency  | 132.31  | 135.82  | 212.73  | 254.16 |
>
> Note that *Avg Decoding* could also be implemented using a speculative-style process, but we adopt standard two-model decoding (i.e., two LLMs generating autoregressively) to emphasize the efficiency gain introduced by speculative decoding in CoSD. In this setting, CoSD is at least **1.5× faster** than baselines that do not use speculative decoding. We will include these results in Table 6 of the revised paper.
>
> > One weakness which sticks out to me is that this method is unlikely to scale well to multiple collaborating LLMs?
>
> We agree that collaborative generation involving more models may reduce the acceptance rate. However, our experiments suggest that the drop is not as drastic as one might expect. Our key observation is that for many common, non-domain-specific token sequences (e.g., “I am”, “This is”), well-trained models tend to produce highly consistent predictions. As shown below, the acceptance rate remains relatively high even as more models are added:
>
> | # of LLMs | Acceptance Rate (%) |
> |----|----|
> | 2   | 81  |
> | 3   | 79  |
> | 4   | 76  |
> | 5   | 77  |
>
> The experiment settings follow Table 6.
>
> Additionally, we can also use a lower number of generated token per step (i.e., lower **K** in Algorithm 1) to drop less tokens, thus ensuring the efficiency when we have a lower acceptance rate. We will clarify and expand upon this discussion in the revised version of the paper.
>
> >From just looking at Fig. 1 it is unclear what the main differences are between CoSD and standard speculative decoding.
>
> We agree with the reviewer that Fig 1 focuses more on the speculative decoding part than the knowledge fusion part. We will modify the figure in the revised paper and emphasize the knowledge fusion ability of the “CoSD Verification” part.
>
> >L138: "generated autoregressively and produced sequentially" is redundant.
>
> Thanks for pointing out the redundant part. We will modify it in the revised paper.
>
> >The results should have error bars... Speculative Decoding should have the same average performance as the assistant model, but exhibits a substantial amount of deviation in the table.
>
> We will add the error bars in the revised paper. The speculative decoding algorithm we use (Miao et al., 2023) has a soft verification strategy, that uses a random number as a threshold to decide whether to accept the draft token for efficiency. In this case, the average performance of speculative decoding will not be exactly the same as the assistant model.
>
> >The assistant model is always "invoked" question.
>
> You're absolutely right that in our current method, the assistant model is always invoked via the speculative decoding. What we intended to convey in the abstract is that the token replacement is invoked when the assistant token differs from the draft model's token and pass the decision rule.
> We agree that the original wording may misleadingly suggest that the assistant model is not used at all unless triggered by a separate process. We will revise this in the final version for clarity, to better reflect the actual mechanism.

---

> > ### Comment · Reviewer_cV7E · 2025-04-08
> >
> > Thanks for the replies to my questions.
> >
> > > On inference speed comparisons
> >
> > Table 6 compares latency between speculative decoding methods only, so this isn't really relevant. Within the framework of speculative decoding, the efficiency will mainly hinge on (a) the acceptance rate and (b) the blockwise parallel benefits of running the verifier, which mainly scales with verifier size.
> >
> > I do not think that naive two-model decoding is the right comparison, since simply producing next token distributions from two independent model can be done in parallel. With some minor synchronization overhead, I would expect that the right baseline is approximately max(token-wise latency model A, token-wise latency model B), not their sum.

---

> > > ### Author Response · Authors · 2025-04-08
> > >
> > > Thanks for the replies and suggestions.
> > >
> > > >I do not think that naive two-model decoding is the right comparison, since simply producing next token distributions from two independent model can be done in parallel. With some minor synchronization overhead, I would expect that the right baseline is approximately max(token-wise latency model A, token-wise latency model B), not their sum.
> > >
> > > Thanks for raising this important point. We would like to clarify that the running time for two-model decoding (even with parallelization) significantly exceeds the maximum of token-wise latency for either model A or model B individually. The main reason is that, during joint token prediction by two models, we cannot effectively utilize inference-time optimizations like KV cache. Specifically, token-wise merging means that the initial predictions from one model might differ from the final chosen token. Consequently, the stored KV cache can become invalid due to incorrect previous tokens, necessitating either frequent recomputation of the entire input or additional mechanisms to detect when KV cache can be reliably used. Our experiments demonstrate that this limitation leads to a substantial increase in running time, often exceeding twice the latency compared to scenarios where KV cache can be smoothly implemented.
> > >
> > > For the experiments we added during the rebuttal, we did make the two models generate in parallel, but the token-wise latency is still more than 3x longer than one model that can utilize KV cache.

---

### Decision · Program_Chairs · 2025-05-01

**Decision:**

Accept (poster)

**Comment:**

This paper proposes to integrate complementary knowledge from multiple LLMs for model knowledge fusion. The proposed method classifies output tokens based on their probabilities to achieve fusion and leverages a speculative decoding framework to enhance generation speed.

In general, the proposed method is intuitive and elegant. Extensive evaluation on 6 LLM pairs and 5 common benchmarks demonstrates the effectiveness of the proposed method, and the major concerns have been addressed by the rebuttal. Therefore, I recommend to accept this paper.